# Effects of iron deficiency and exogenous sucrose on the intermediates of chlorophyll biosynthesis in *Malus halliana*

**Aixia Guo**[1,2], **Ya Hu**[1,2], **Mingfu Shi**[1], **Hai Wang**[1], **Yuxia Wu**[1], **Yanxiu Wang**[1] *

**1** College of Horticulture, Gansu Agricultural University, Lanzhou, Gansu, China, **2** Northwest Institute of Eco-Environment and Resources, Chinese Academy of Science, Lanzhou, Gansu, China

* wangxy@gsau.edu.cn

**Data Availability Statement:** All relevant data are within the manuscript and its Supporting Information files.

## Abstract

*Malus halliana* is an iron (Fe)-efficient apple rootstock growing in calcareous soil that shows obvious 'greenness' traits during Fe deficiency. Recent studies have shown that exogenous sugars can be involved in abiotic stress. To identify the key regulatory steps of chlorophyll (Chl) biosynthesis in *M. halliana* under Fe deficiency and to verify whether exogenous sucrose (Suc) is involved in Fe deficiency stress, we determined the contents of the Chl precursor and the expression of several Chl biosynthetic genes in *M. halliana*. The results showed that Fe deficiency caused a significant increase in the contents of protoporphyrin IX (Proto IX), Mg-protoporphyrin IX (Mg-Proto IX) and protochlorophyllide (Pchlide) in *M. halliana* compared to the Fe-sensitive rootstock *Malus hupehensis*. Quantitative real-time PCR (RT-qPCR) also showed that the expression of protoporphyrinogen oxidase (*PPOX*), which synthesizes Proto IX, was upregulated in *M. halliana* and downregulated in *M. hupehensis* under Fe deficiency. Exogenous Suc application prominently enhanced the contents of porphobilinogen (PBG) and the subsequent precursor, whereas it decreased the level of δ-aminolaevulinic acid (ALA), suggesting that the transformation from ALA to PBG was catalyzed in *M. halliana*. Additionally, the transcript level of δ-aminolevulinate acid dehydratase (*ALAD*) was noticeably upregulated after exogenous Suc treatment. This result, combined with the precursor contents, indicated that Suc accelerated the steps of Chl biosynthesis by modulating the *ALAD* gene. Therefore, we conclude that *PPOX* is the key regulatory gene of *M. halliana* in response to Fe deficiency. Exogenous Suc enhances *M. halliana* tolerance to Fe deficiency stress by regulating Chl biosynthesis.

## Introduction

Iron (Fe) is an essential micronutrient required by all plants [1]. Fe deficiency in plants results in severe chlorosis of leaves [2]. The application of tolerant rootstocks is an effective method to prevent chlorosis in fruit production due to Fe stress [3]. *Malus halliana*, an indigenous apple rootstock originating from arid saline-alkali habitats in Gansu, grows very well and shows characteristics of a Fe deficiency-tolerant rootstock. The chlorosis associated with Fe deficiency was not found in the northwest Loess Plateau of China [4].

**Funding:** This work was supported by Gansu Agricultural University Youth Postgraduate Tutor Support Fund Project (project No. GAU-2NDS-201710).

**Competing interests:** The authors declare that they have no conflict of interest.

**Abbreviations:** Fe, Iron; M. halliana, *Malus Halliana*; M. hupehensis, *Malus hupehensis*; Chl, Chlorophyll; Suc, Sucrose; ALA, δ-aminolaevulinic acid; PBG, Porphobilinogen; URO III, Uroorphyinogen III; Proto IX, Protoporphyrin IX; Mg-Proto IX, Mg-protoporphyrin IX; Pchlide, Protochlorophyllide; ALAD, δ- aminolevulinate acid dehydratase; PBGD, Porphobilinogen deaminase; UROS, Uroporphyrinogen III synthase; UROD, Uroporphyrinogen decarboxylase; PPOX, Protoporphyrinogen IX oxidase; MgCh, Magnesium chelatase; MgPMT, Magnesium-protoporphyrin IX methyltransferase; MPE, Mg-protoporphyrin IX monomethyl ester cyclase; RT-qPCR, Quantitative Real-time PCR.

Leaf chlorosis may be caused by deficient chlorophyll (Chl) biosynthesis [5]. Chl biosynthesis plays essential roles in photosynthesis and plant growth in response to environmental change [6]. The Chl biosynthesis pathway has many steps and involves various enzymes, and a blockade in one step will affect Chl biosynthesis and cause changes in leaf color [7]. The key regulatory sites of Chl synthesis are different for each crop under external stress. A study of adzuki bean reported that the transformation of protoporphyrin IX (Proto IX) is blocked in Chl synthesis, causing etiolated seedlings [8]. Salinity-alkalinity stress disrupted Chl synthesis by blocking the conversion of URO III to Proto IX, which reduced the Chl content in tomato [9]. Another study suggested that Chl biosynthesis is blocked in a mutant at the Chl a production step, and the expression of multiple genes related to Chl biosynthesis was downregulated in *pylm* [10]. A study on the effect of different light qualities of LEDs on the Chl biosynthesis precursors of nonheading Chinese cabbage showed that red plus blue LEDs enhanced Chl biosynthesis precursors [11]. Remarkably, Fe deficiency directly affected Chl synthesis [12]. Research in poplar revealed that Chl synthesis was inhibited under Fe-deficient conditions [13]. Spiller et al. attempted to study the effect of Fe on the Chl biosynthetic pathway, and the results indicated that Fe deficiency leads to the accumulation and excretion of intermediates in the tetrapyrrole biosynthetic pathway, particularly coproporphyrin [14]. Moreover, an investigation was initiated to locate possible sites where a deficiency of Fe might limit Chl synthesis of cowpea plants [2]. However, the responses of Chl biosynthesis to Fe deficiency stress and the key regulatory sites in *M. halliana* are still unknown.

Sucrose (Suc) is the major sugar that plants assimilate in photosynthesis and transport to various nonphotosynthetic tissues; it was not only originally recognized as an energy source for metabolism but also functions as a signaling molecule involved in the regulation of various physiological processes in plants [15–18]. Increased accumulation of Suc is a critical requirement for the adaptation of plants to stresses [19, 20]. Higher accumulation of soluble sugars in roots increases the resistance of maize plants to salt-induced osmotic stress [21]. Increased Suc accumulation is required for the regulation of Fe deficiency responses in *Arabidopsis* plants [22]. However, little is known about how exogenous Suc regulates the response of *M. halliana* to Fe deficiency through Chl synthesis.

Studies of Chl biosynthesis have focused on various aspects [23, 24], whether biochemical [25, 26] or genetic [27, 28]. However, gaps remain in the knowledge of Chl biosynthesis and the related genes in apple rootstocks. Therefore, it is important to elucidate the Chl biosynthetic molecular responses of *M. halliana* to Fe deficiency. In this article, we characterized the Chl biosynthetic pathway and gene expression patterns of protoporphyrin IX and porphobilinogen precursor formation in *M. halliana* under Fe deficiency and exogenous Suc. This study provides a foundation for improved understanding of Fe tolerance responses in apples and gives insights into the functional characterization of Fe resistance genes.

## Materials and methods

### Plant materials and treatment of iron deficiency

Seeds of *M. halliana* and *M. hupehensis* (provided by Lanzhou, Gansu Province, China) were surface-sterilized in 0.2% $KMnO_4$ for 30 min and then washed with running water for 12 h. Seeds were subsequently stratified at 4°C sand for 40 d, and germinated seeds were directly sown into plastic pots filled with substrates. Seedlings with eight true leaves were employed as test materials. Uniform seedlings were transferred to foam boxes filled with half-strength Han's nutrient solution [29] for preculture. The nutrient solution was aerated and renewed every 7 d. After 14 d, the seedlings were transferred to Han's nutrient solution that contained either 4 μM (-Fe) or 40 μM Fe (III)-EDTA (CK). After 0, 0.5, 3, 6 and 12 d of Fe deficiency, the

leaves were used to measure Chl precursors. However, according to transcriptome data analysis [30], the relative expression of related genes was assayed after 0, 0.5 and 3 d of Fe deficiency.

## Exogenous sucrose treatments

Eight-true-leaf *M. halliana* seedlings were uniformly transferred to foam boxes with half-strength Han's nutrient solution for 14 d to adapt to the environment for hydroponic cultivation. After 14 d of Fe deficiency, the uniform seedlings were transferred to Han's nutrient solution containing 4 μM Fe, 40 μM Fe, 2 mM Suc [22] and 40 μM Fe mixed with 2 mM Suc. Therefore, this part of the experiment included six treatments: T1 (-Fe 0 d), T2 (-Fe 14 d), T3 (-Fe 21d), T4 (-Fe 14 d +Fe 7 d), T5 (-Fe 14 d +Suc 7 d), T6 (-Fe 14 d + (Fe +Suc) 7 d). Leaf samples were collected at different times, frozen immediately in liquid nitrogen, and stored in the refrigerator at -80˚C until needed.

## Determination of chlorophyll precursors

δ-aminolaevulinic acid (ALA) was determined as described by Morton [31]. Porphobilinogen (PBG) and Uroorphyinogen III (URO III) were measured according to Bogorad [32]. Protoporphyrin IX (Proto IX), Mg-protoporphyrin IX (Mg-Proto IX), and Protochlorophyllide (Pchlide) were assayed following the method of Hodgins and Van Huystee [33].

## RNA isolation and real-time PCR

Total RNA was isolated using a TRIzol kit (Invitrogen, Carlsbad, CA, USA). After extraction, the RNA quality was measured by gel electrophoresis using 1% agarose gel. For real-time PCR (RT-qPCR) analysis, cDNA was synthesized from total RNA using the PrimeScript™ RT reagent Kit with gDNA Eraser (Perfect Real Time) (TaKaRa, Dalian, China) according to the manufacturer's instructions. RT-qPCR was performed with a DNA Engine Opticon System using a Light Cycler® 96 Instrument (Roche, Shanghai, China). *GAPDH* was used as a reference gene. Measurements for each plate were replicated three times. The relative gene expression levels were calculated using the $2^{-\Delta\Delta Ct}$ method. RT-qPCR primer pairs are tabulated in Table 1.

## Statistical analysis

All data were analyzed with SPSS version 22.0 for Windows (IBM, Armonk, NY, USA). Graphs were generated using the Origin 9.0 software (Origin Lab, Hampton, MA, USA). The results are presented as the means of three independent experiments.

Statistical analysis of parameters was tested by analysis of variance and mean comparison was performed with a Duncan's test ($P < 0.05$)

# Results

## Effects of iron deficiency on the chlorophyll precursor contents of *M. halliana* and *M. hupehensis*

As shown in Fig 1, the contents of six precursors in *M. halliana* were noticeably higher than those in *M. hupehensis*. From 0 to 12 d of Fe deficiency, the contents of ALA, PBG and URO III of the two apple rootstocks increased (Fig 1A, 1B and 1C). However, the levels of Proto IX, Mg-Proto IX, and Pchlide in *M. halliana* were generally increased but decreased in *M. hupehensis*. Moreover, the levels in *M. halliana* were approximately 2-fold those in *M. hupehensis*

**Table 1. Primers used for RT-qPCR.**

| Gene name | Gene ID | Primer sequence (5'-3') | |
| --- | --- | --- | --- |
| | | Forward primer | Reverse primer |
| ALAD | 103423306 | GCGTTGTCATGGAGTCCTGATGG | CCAGTTGGCGACCACTTCAGC |
| PBGD | 103423833 | CCTTGCAACCTTCCGCGAGAG | TCAGCCGTGTCTGGACGTTACC |
| UROS | 103455996 | CCACCTTCTTGTCCGCCACTTC | TTGCTGTTCTTGCCGTGCTCTC |
| UROD | 103444879 | AGGTAGAAGGCGACTGGGAC | CCCTCTACCGGCTTTCCTCA |
| PPOX | 103444480 | TTCTGTTGACTGCGTGGTGGTG | GGTGTCGCCGTGCTTGGTAG |
| CHLH | 103456287 | AATACCAAAGCCTAACTCC | AACAGCAGCCTCATCG |
| MgPMT | 103454888 | AAAACCTACCACCCTAAA | CTTCACCACCTCCTTGT |
| MPE | 103454703 | CTTTGCTCTGCGTTGT | GCTGTGGTGCGATTT |

(Fig 1D, 1E and 1F). Therefore, we hypothesized that the regulatory site of Chl biosynthesis in *M. halliana* affected Fe deficiency was the transformation from URO III to Proto IX.

## Expression of genes involved in chlorophyll biosynthesis under iron deficiency

To prove the validity of the above hypothesis and gain insight into the key genes of *M. halliana* in response to Fe deficiency, we determined the relative expression levels of 8 genes related to Chl synthesis under Fe deficiency. Fig 2 shows that the relative levels of genes in *M. halliana* were dramatically higher than those in *M. hupehensis*. The relative levels of the δ- aminolevulinate acid dehydratase (*ALAD*), porphobilinogen deaminase (*PBGD*) and uroporphyrinogen III synthase (*UROS*) genes in *M. halliana* and *M. hupehensis* increased first and then dropped, peaking on 0.5 d, and the expression was no more than 3 in both groups. (Fig 2A, 2B and 2C). Thereafter, at 0.5 d, the relative levels of Uroporphyrinogen decarboxylase (*UROD*) and Protoporphyrinogen IX oxidase (*PPOX*) in *M. halliana* increased quickly, especially the expression of *PPOX*, which was approximately 75 in *M. halliana* instead of no more than 1 in *M. hupehensis* (Fig 2D and 2E). These results suggested that *PPOX* was responsive mainly to stresses and could improve resistance to Fe deficiency. Subsequently, the relative expression of genes in *M. halliana*, such as Magnesium chelatase (Subunit *CHLH*), Magnesium-protoporphyrin IX methyltransferase (*MgPMT*), and Mg-protoporphyrin IX monomethyl ester cyclase (*MPEs*), exhibited a gradually increasing trend during stress at 0–3 d (Fig 2F, 2G and 2H). These results were consistent with the Chl precursor analysis.

## Regulation of the iron deficiency phenotype by exogenous sucrose

We found that the leaves of *M. halliana* initially showed chlorosis symptoms after 12 d of Fe deficiency, and one of the important reasons was believed to be the loss of Suc or Fe. To investigate the effects of exogenous Suc on the phenotype of *M. halliana* seedlings, we conducted phenotype analysis of plants grown hydroponically in Suc-and Fe-sufficient and Fe-deficient conditions (Fig 3). The phenotype revealed that differences still existed. After 21 d of growth in Fe-deficient conditions (T3), the leaves of the plants exhibited more severe chlorosis symptoms than those of the plants treated with T4, T5, and T6. Additionally, the plants supplied with Suc and Fe (T6) showed pale green leaves compared with those under the T4 and T5 treatments. The Suc-supplied leaves (T5) were greener than the Fe-supplied leaves (T4). As a result, leaf chlorosis was ameliorated by Suc application. These results showed that the tolerance of Fe deficiency of *M. halliana* was positively regulated by Suc.

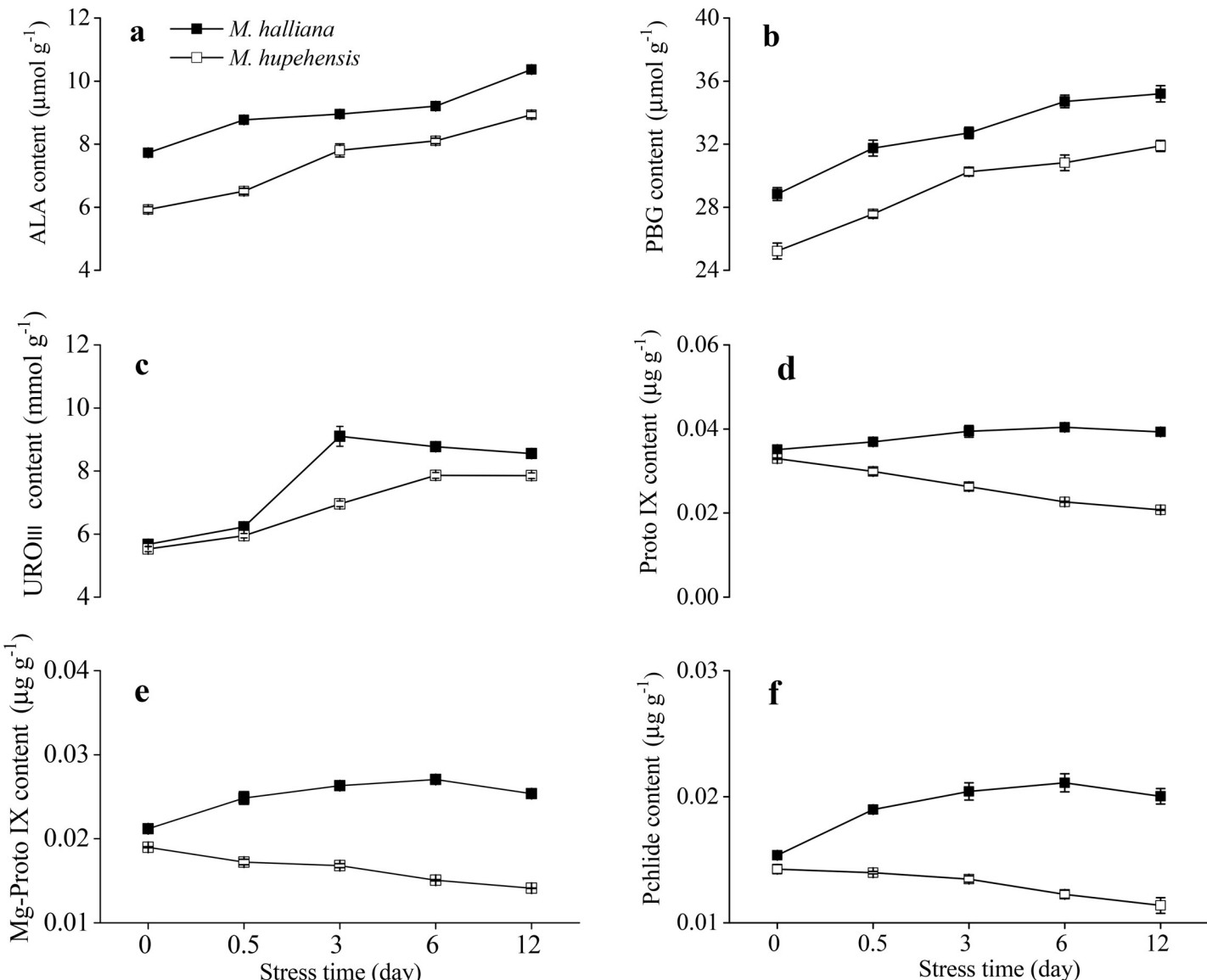

**Fig 1. Effects of Fe deficiency on Chl precursor contents in the leaves of *M. halliana* and *M. hupehensis*.** The seedlings underwent Fe deficiency for 0, 0.5, 3, 6, and 12 d as indicated. (a) ALA content. (b) PBG content. (c) URO III content; (d) Proto IX content. (e): Mg-Proto IX content. (f): Pchlide content. Vertical bars represent the mean ± SD value from three temporal replicates (n = 3).

### Effects of exogenous sucrose on the chlorophyll precursor contents in *M. halliana*

As shown in Fig 3, exogenous Suc may also be required for regulating the tolerance of Fe deficiency in *M. halliana*. Chlorosis was attributable to the reduction of Chl content causing defects in Chl biosynthesis. We determined the contents of the Chl precursors of the six treatments. Compared with the T3 treatment, the T5 treatment resulted in a clear decrease in the ALA level (Fig 4A) but increased the contents of PBG and URO III. (Fig 4B and 4C). The Chl intermediates Mg-Proto IX and Pchlide under T5 treatment were not significantly different from those under the other treatments (Fig 4D, 4E and 4F). The transformation of ALA to PBG and PBG to URO III was enhanced after application of Suc. As a result, after the

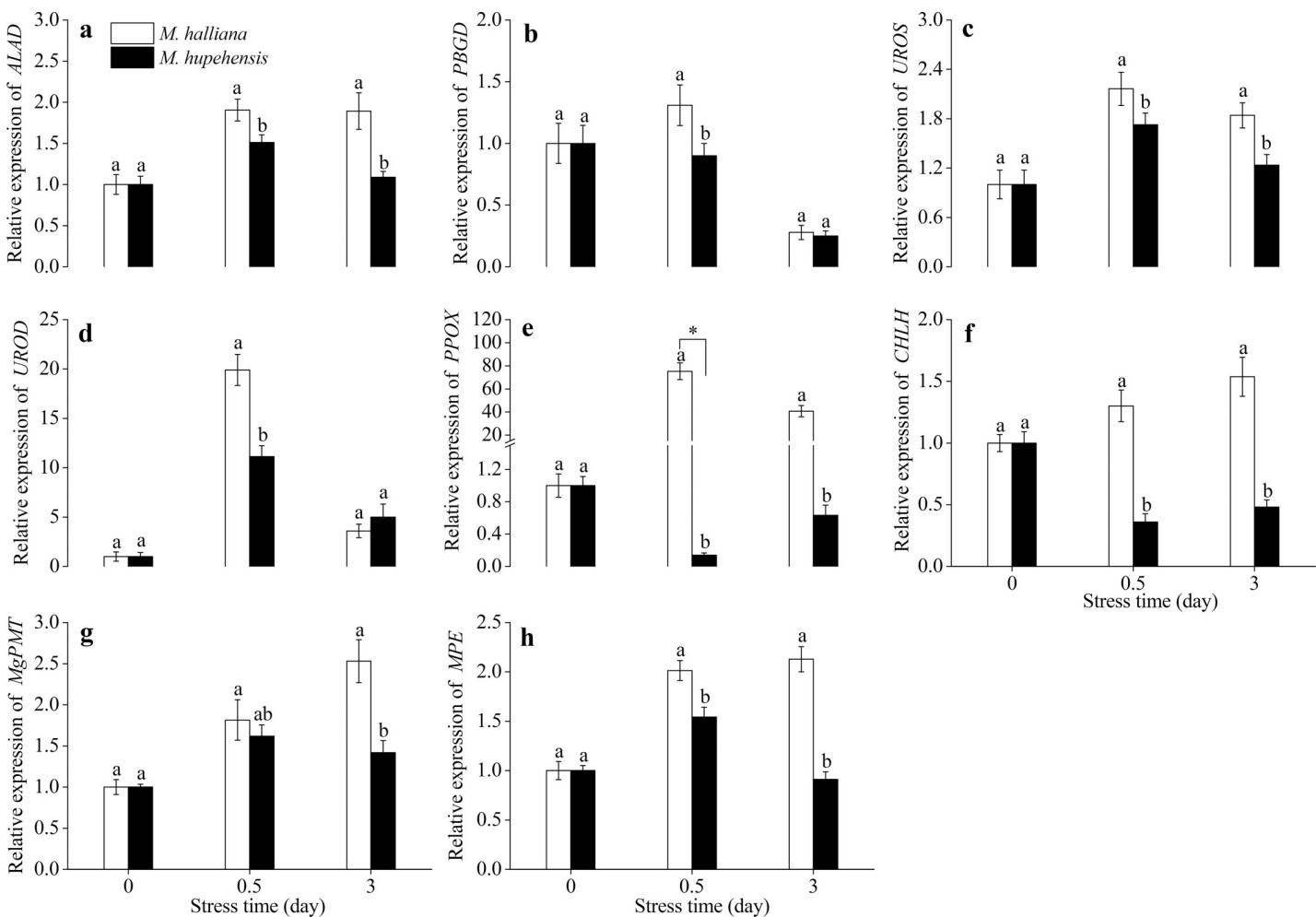

**Fig 2. Expression pattern of eight genes in response to Fe deficiency in *M. halliana* and *M. hupehensis*.** (a)Expression of *ALAD*. (b) Expression of *PBGD*. (c) Expression of *UROS*. (d) Expression of *UROD*. (e) Expression of *PPOX*. (f) Expression of *CHLH*. (g) Expression of *MgPMT*. (h) Expression of *MPE*. Data are expressed as the mean ± SD (n = 3). The different letters and an asterisk show significant differences between the two apple rootstocks at each time point ($P < 0.05$).

application of exogenous Suc, the regulatory step of Chl precursor synthesis in *M. halliana* affected by Fe deficiency might be the conversion of ALA to PBG, suggesting this process was enhanced.

## Comparison of gene expression in *M. halliana* after sucrose application

Given that exogenous Suc catalyzed the steps of Chl biosynthesis, we wanted to test the effects of exogenous Suc on the relative expression levels of related genes. As shown in Fig 5, compared with the other treatments, T5 significantly enhanced the expression of *ALAD*, *PBGD* and *UROS* (Fig 5A, 5B and 5C), consistent with a change in the corresponding precursor contents (Fig 4). Therefore, the expression of *ALAD* was significantly upregulated by the Suc treatment under Fe-deficient conditions. Nevertheless, for the other five genes, their expression levels not affected by the application of exogenous Suc, Fe or Suc and Fe. Enhanced relative levels of *UROD*, *PPOX*, *CHLH*, *MgPMT*, and *MPE* were associated with Fe deficiency, peaking on 21 d of Fe deficiency. Therefore, Fe deficiency caused a dramatic elevation in the relative levels of the above genes.

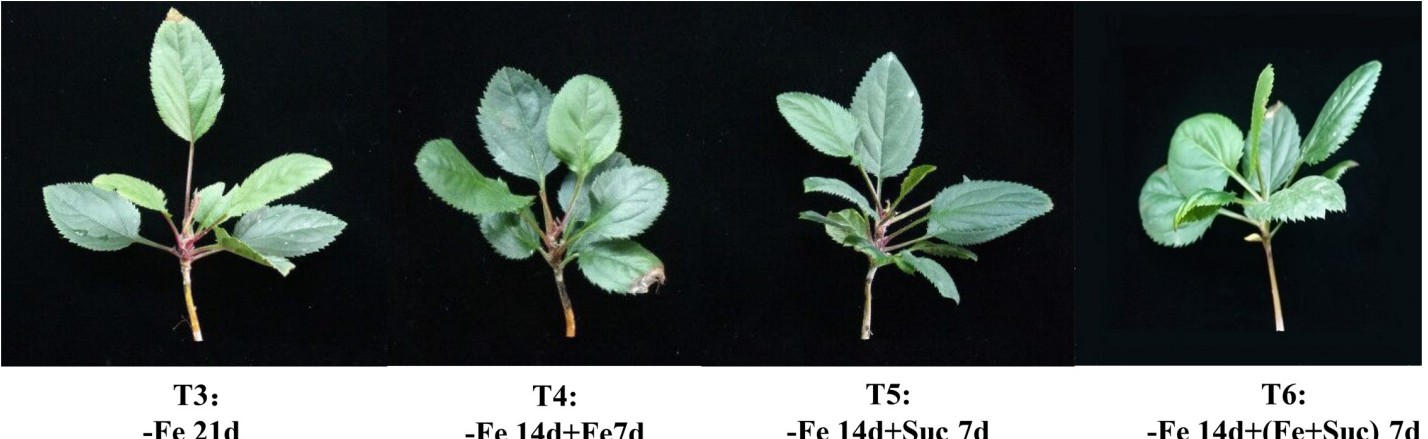

| T3:<br>-Fe 21d | T4:<br>-Fe 14d+Fe7d | T5:<br>-Fe 14d+Suc 7d | T6:<br>-Fe 14d+(Fe+Suc) 7d |

**Fig 3. Effects of exogenous Suc on the phenotype of leaves from *M. halliana*.** T3: -Fe 21 d (Fe deficiency 21 d). T4: -Fe 14 d + Fe 7 d (applied Fe after 14 d of Fe deficiency). T5: -Fe 14 d + Suc 7 d (applied Suc after 14 d of Fe deficiency). T6: -Fe 14 d+ (Fe + Suc) 7 d (applied Fe and Suc after 14 d of Fe deficiency).

### Schematic model of key genes involved in chlorophyll biosynthesis during iron deficiency and sucrose application

Overall, based on the variations in Chl precursor contents (Figs 1 and 5), gene expression (Figs 2 and 6) and phenotype characterization (Fig 3), we concluded that, under short-term stress, the key for the 'greenness' of *M. halliana* was high expression of *PPOX*. After long-term stress (over 12 d), *M. halliana* showed chlorosis symptoms.

Exogenous Suc can alleviate chlorosis. The accumulation of PBG precursor under Suc treatment enhanced the Chl synthesis, and the corresponding gene *ALAD* was up-regulated. Hence, *PPOX* and *ALAD* were key genes in mediating Fe-deficient and exogenous Suc-regulated Chl biosynthesis, respectively (Fig 6). Application of Suc could enhance the tolerance of *M. halliana* to Fe deficiency through improvement in Chl biosynthesis.

## Discussion

Iron is an essential element for all living organisms, functioning in various cellular processes, such as Chl biosynthesis [34]. Fe deficiency is a nutritional disorder in plants, contributing to chlorosis by limiting Chl biosynthesis [35]. Disruption of any one step of Chl biosynthesis may lead to evident accumulation of the intermediates produced in previous steps, leading to disruption and substantial decreases in the amount of products produced in the subsequent steps. Previous research has found that seawater stress hinders the transformation of PBG to URO III in spinach [36]. The study showed that UV-B disrupts Chl synthesis at the point of ALA conversion to PBG [37]. In the present study, Chl synthesis at the step of URO III conversion to Proto IX indicated that Fe deficiency stress disrupted Chl biosynthesis in *M. hupehensis*, and the Chl biosynthesis of *M. halliana* was not blocked after Fe deficiency. The leaves of *M. hupehensis* exhibited chlorotic symptoms before *M. halliana*, consistent with the finding that the Chl precursor contents of *M. hupehensis* were significantly lower than those of *M. halliana*. Similar to a study on cabbage, *M. halliana* could maintain plant growth and preserve adequate chlorophyll synthesis under iron-limiting conditions, probably due to its better Fe-use efficiency than *M. hupehensis* [38]. The accumulation of chlorophyll intermediate metabolites can sometimes prevent adverse effects on *M. halliana*.

Protoporphyrinogen IX oxidase is the last enzyme in the common pathway of heme and chlorophyll synthesis and catalyzes the oxidation of protoporphyrinogen-IX to protoporphyrin

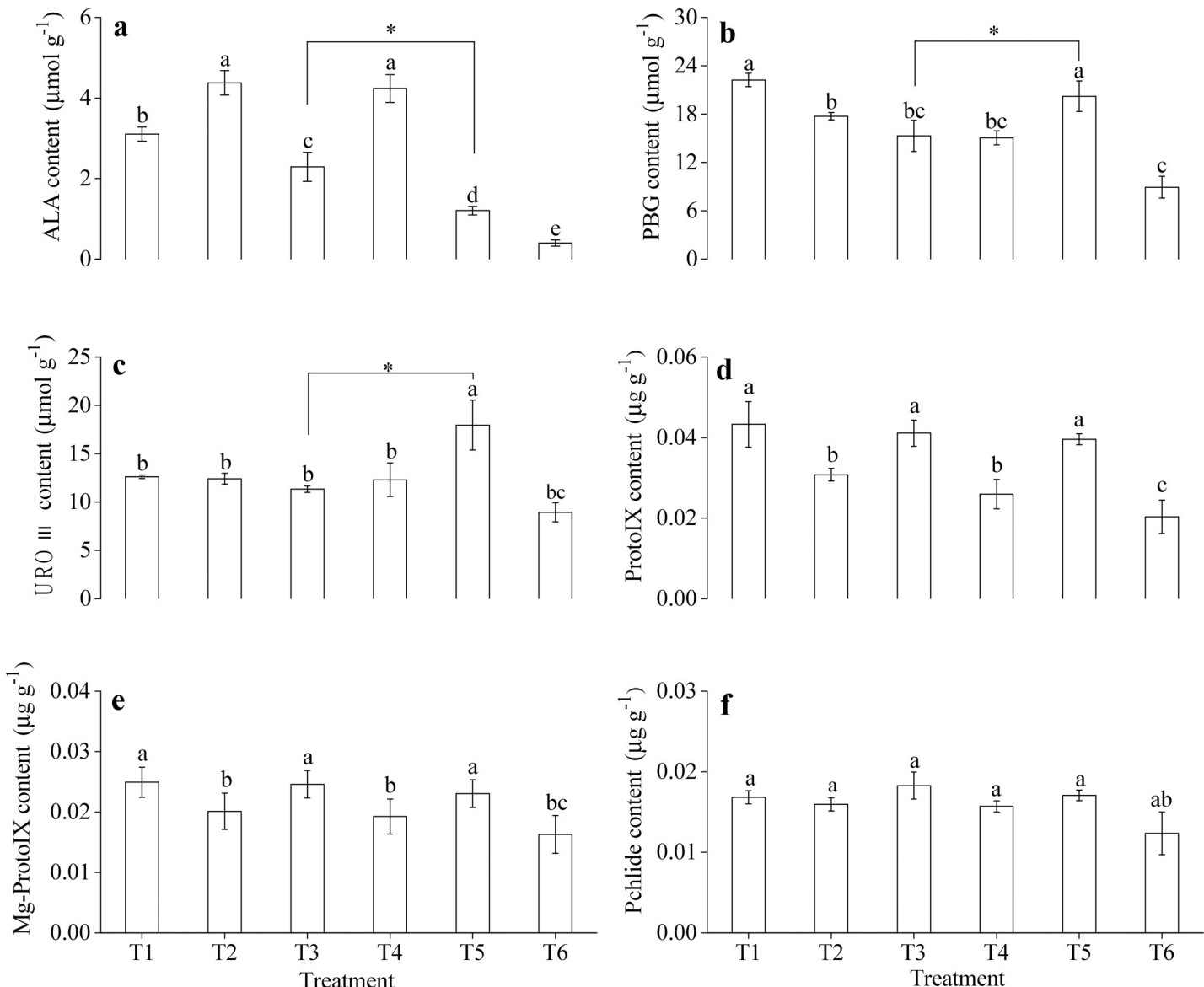

**Fig 4. Effects of exogenous Suc on Chl precursor contents in the leaves of *M. halliana*.** (a) ALA content. (b) PBG content. (c) URO III content. (d) Proto IX content. (e) Mg- Proto IX content. (f) Pchlide content. Data are the mean ± SD (n = 3). Different letters indicate significant differences among treatments ($P < 0.05$).

IX by molecular oxygen [39, 40]. Research has suggested that the *PbPPO1* gene might be involved in core browning under modified atmosphere storage in 'Yali' pears [41]. The *PPO* gene expression level in response to *Aspergillus tubingensis* in table grapes was enhanced with trehalose [42]. In this study, the relative expression of *PPOX* changed slightly in *M. hupehensis* but showed high expression levels in *M. halliana*, indicating that *PPOX* was the key gene in response to Fe deficiency in *M. halliana*, which lays the foundation for cloning genes responsive to Fe deficiency stress.

Sucrose plays a vital role in plant growth and development as well as the response to abiotic stress [43]. The application of sucrose in unripe strawberries resulted in the induction of ripening [44]. A study found that sucrose is one of most abundant metabolites in the glucose metabolic pathway, which plays an indispensable role in balancing photosynthetic activity in *M.*

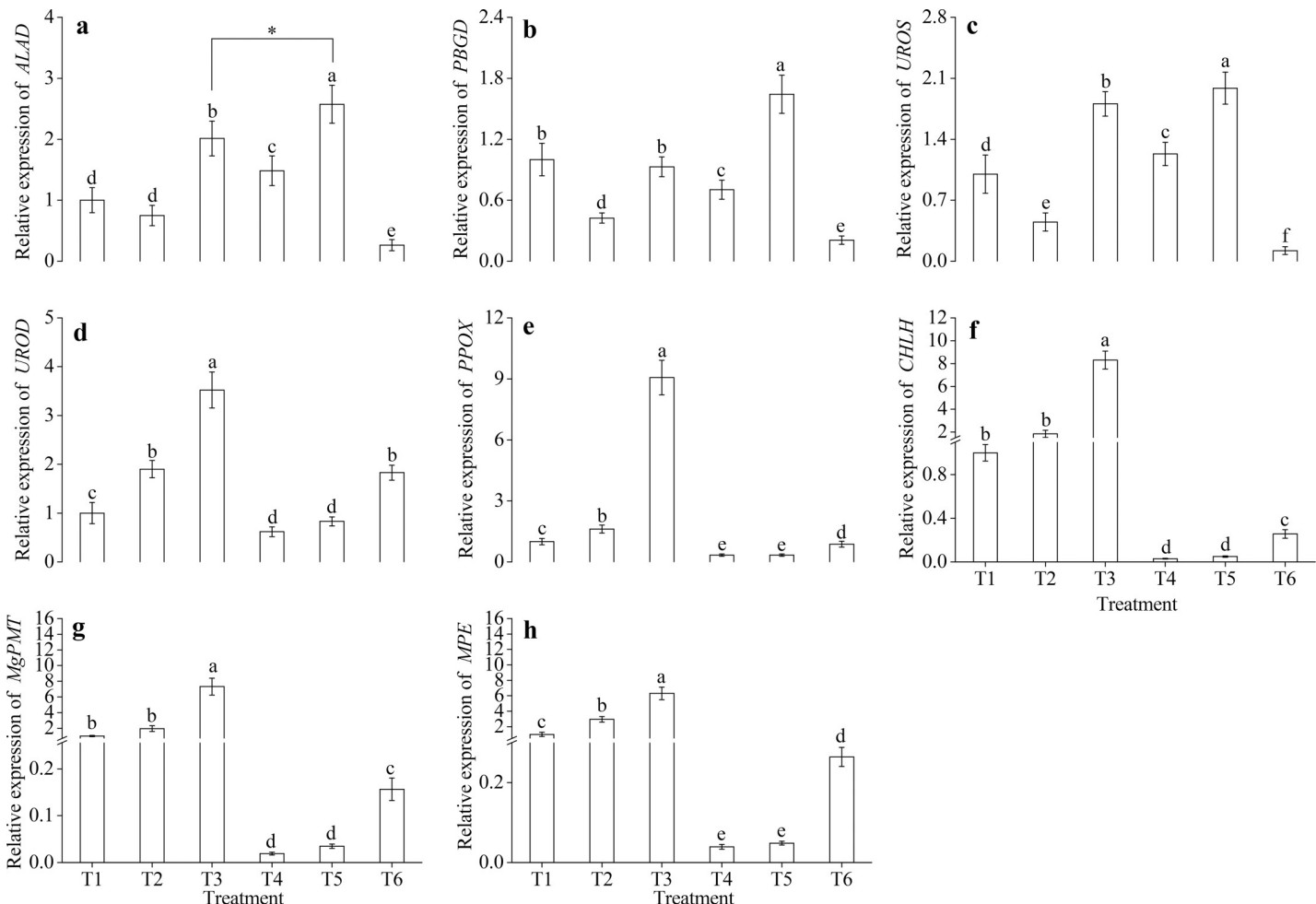

**Fig 5. Comparison of gene expression in the leaves of *M. halliana* under exogenous sucrose treatment.** (a) Expression of *ALAD*. (b) Expression of *PBGD*. (c) Expression of *UROS*. (d) Expression of *UROD*. (e) Expression of *PPOX*. (f) Expression of *CHLH*. (g) Expression of *MgPMT*. (h) Expression of *MPE*. Data are expressed as the mean ± SD (n = 3). Different letters indicate significant differences between 6 treatments (*P* < 0.05).

*halliana* [45]. In this study, Fe-deficient apple seedlings showed a typical Fe-stress phenotype, becoming yellow. The phenotypes of plants treated with Suc, Fe, and both Suc and Fe were found to be similar and showed good growth, and the Suc-supplied plants were greener than the Fe-supplied and Fe-Suc-supplied plants. This phenotype indicated that the repressions of Fe deficiency was partially reversed by exogenous Suc application. Furthermore, exogenous Suc increased the contents of PBG and subsequent precursors but decreased the ALA content in *M. halliana*, suggesting that the key regulatory point of Chl biosynthesis was moved forward after the application of exogenous Suc. Therefore, Suc was involved in the regulation of Chl intermediate products in *M. halliana* under Fe deficiency.

δ-Aminoleuvulinate acid dehydratase catalyzes the formation of PBG from two ALA molecules via the formation of two successive Schiff base intermediates [46, 47]. The *ALAD* gene, encoding δ-aminolevulinic acid dehydratase, is the rate-limiting enzyme of Chl biosynthesis [48]. In this study, after exogenous Suc application, increased expression of *ALAD* in *M. halliana* suggested that the accumulated intermediates of the PBG may have been efficiently utilized and enhanced the Chl biosynthetic pathway. As a result, in response to Fe deficiency, Suc

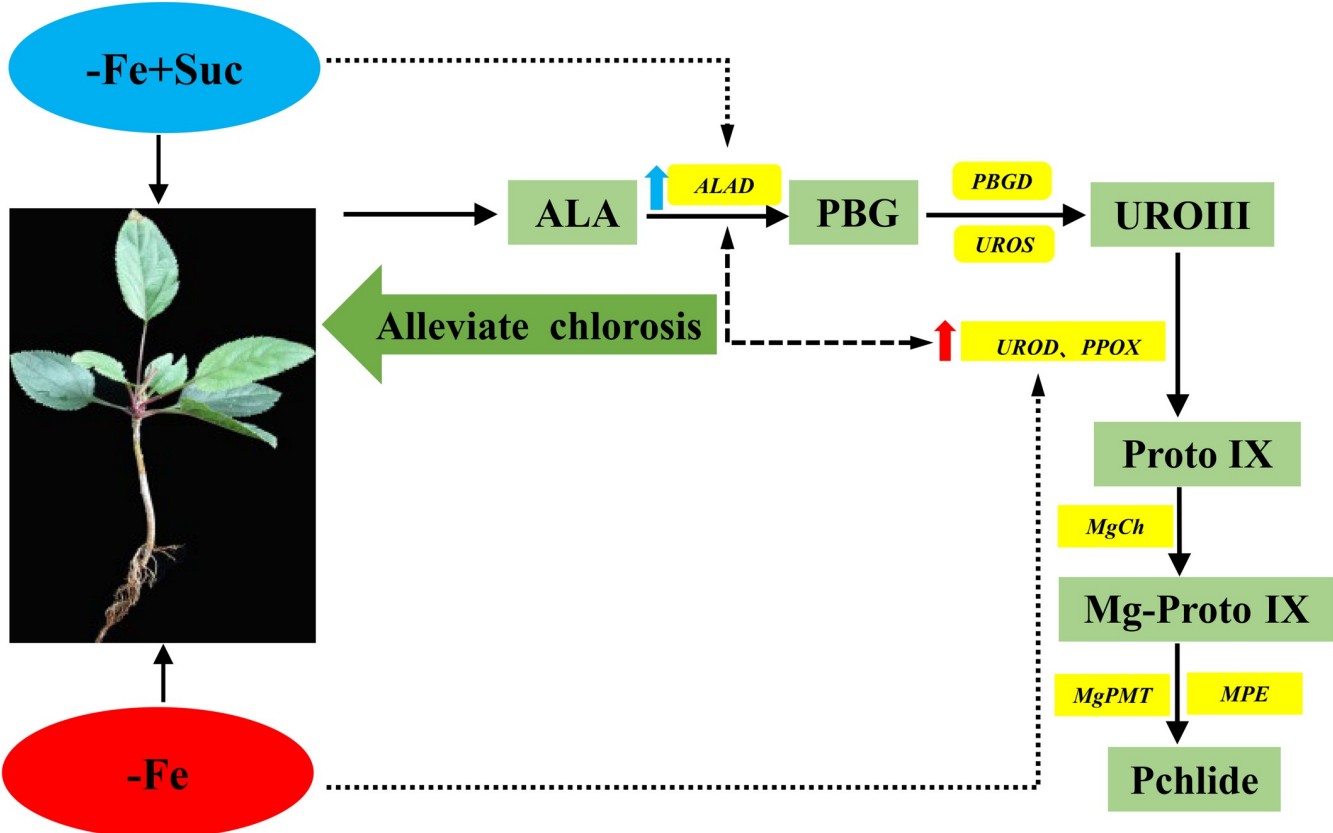

**Fig 6. The key synthetic sites of Chl biosynthesis under Fe deficiency and exogenous Suc treatment.** Results showed that (1) *PPOX* expression of *M. halliana* was enhanced to adapt to Fe-deficient condition. (2) Under Fe deficiency, application of exogenous Suc increased the expression of *ALAD* and further enhanced Chl synthesis.

may act as a signaling molecule to regulate the upregulation of the expression of related genes [49–51].

## Conclusion

Our experimental results have showed that up regulation of *PPOX* gene enhanced Chl biosynthesis of *M. halliana*. Suc positively regulated the responses to Fe deficiency in *M. halliana* via Chl biosynthesis. Thus, *PPOX* is the key regulatory gene of *M. halliana* in response to Fe deficiency. Exogenous Suc application on apple seedlings could ameliorate the adverse effects caused by Fe deficiency. In future work, cloning and functional characterization of the key genes for Chl biosynthesis of *M. halliana* in response to Fe deficiency will be performed.

## Supporting information

**S1 File. Relevant data underlying the finding described in manuscript.**
(XLSX)

## Author Contributions

**Data curation:** Ya Hu, Mingfu Shi.

**Supervision:** Yanxiu Wang.

**Writing – original draft:** Aixia Guo.

**Writing – review & editing:** Hai Wang, Yuxia Wu.

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
