## [Decision Letter · Decision Letter 0]

18 Feb 2020

PONE-D-19-35123

Effects of Iron Deficiency and Exogenous Sucrose on the Intermediates of Chlorophyll Biosynthesis in Malus halliana

PLOS ONE

Dear Dr Wang,

Thank you for submitting your manuscript to PLOS ONE. After careful consideration, we feel that it has merit but does not fully meet PLOS ONE’s publication criteria as it currently stands. Therefore, we invite you to submit a revised version of the manuscript that addresses the points raised during the review process.

ACADEMIC EDITOR: Thank you for submitting your manuscript to PLOS ONE. I agree with the reviewers that English language needs to be checked thoroughly. Main claims of the paper are not properly placed in the context of previous literature. Authors did not treat the literature fairly. Introduction is too general. 

We would appreciate receiving your revised manuscript by 45 days. To enhance the reproducibility of your results, we recommend that if applicable you deposit your laboratory protocols in protocols.io, where a protocol can be assigned its own identifier (DOI) such that it can be cited independently in the future. For instructions see: http://journals.plos.org/plosone/s/submission-guidelines#loc-laboratory-protocols

We look forward to receiving your revised manuscript.

Kind regards,

Basharat Ali, Ph.D

Academic Editor

PLOS ONE

Additional Editor Comments (if provided):

Thank you for submitting your manuscript to PLOS ONE. I agree with the reviewers that English language needs to be checked thoroughly. Main claims of the paper are not properly placed in the context of previous literature. Authors did not treat the literature fairly. Introduction is too general.

Journal Requirements:

https://bmcplantbiol.biomedcentral.com/articles/10.1186/s12870-015-0699-7

In your revision ensure you cite all your sources (including your own works), and quote or rephrase any duplicated text outside the methods section. Further consideration is dependent on these concerns being addressed.

3. In your Methods, please state the source of the seeds used in your study.

Reviewers' comments:

Reviewer's Responses to Questions

**Comments to the Author**

1. Is the manuscript technically sound, and do the data support the conclusions?

Reviewer #1: Yes

Reviewer #2: No

2. Has the statistical analysis been performed appropriately and rigorously? 

Reviewer #1: Yes

Reviewer #2: No

3. Have the authors made all data underlying the findings in their manuscript fully available?

Reviewer #1: Yes

Reviewer #2: Yes

4. Is the manuscript presented in an intelligible fashion and written in standard English?

Reviewer #1: Yes

Reviewer #2: No

5. Review Comments to the Author

Reviewer #1: Apple is an important fruit due to its nutritional and commercial value and study of apple rootstocks is an important research direction. It is imperative to fully understand the complex mechanisms involved under stressful environments. The authors presented the effects of Iron deficiency and exogenous application of Sucrose on the intermediates of chlorophyll biosynthesis in Apple rootstocks. The results are interesting and valuable for scientific research but I have several concerns. The manuscript is not acceptable in its present form. If authors can address the stress modulated mechanism in more elaborated way then the paper is worth of publication after major revision.

These points must be addressed:

1. First you need to mention the common name in Title. You can mention scientific name in Abstract or Introduction in Parenthesis. Each scientific name has “L.” at the end according to nomenclature therefore, revise it thoroughly.

2. All abbreviations must be mentioned in full form at its first place. Mention all these abbreviations in footnotes of the first page.

3. In the last paragraph of Introduction you have described some results. You just need to mention the background of your work as well as the significance for scientific world. You should discuss results in discussion part.

4. Insert line numbers in the manuscript for better review process.

5. Meaning is not clear “pylm”….is it Palm??

6. “nonphotosynthetic must be replaced with non-photosynthetic”

7. You have mentioned “Increased Suc accumulation is required for regulating Fe-deficiency responses in plants, with auxins acting downstream in transmitting the Fe-deficiency signal”. How could you explain the downstreaming??

8. “Exogenous Suc treatments”. Must mention Sucrose in full form in heading or subheadings. Similarly “Determination of Chl Precursors” as Determination of Chlorophyll Precursors.

9. Caption of Fig 3. shows treatment codes as T1, T2 etc….all the punctuation must be appropriate like T3: Fe 21 d. Revise all codes in figure captions.

10. You have used various abbreviations in your manuscript. You should also include Pn for Photosynthesis.

11. “Stay green” must be replaced with greenness.

12. You must have to mention your main findings/conclusions under separate heading. Also mention your major outcomes in more elaborative way.

13. Arrange the references alphabetically.

14. For the figures, you need to mention the x axis once for A, C, E and combine them, then mention x-axis scale once for B, D and F and combine them. Move Y-axis to right side of three figures (B, D, F). Similarly, move Y-axis on the left side of three figures (A, C, E). Combine all the six figures into single figure and enhance the dpi up to 500 for better understanding. Follow this procedure for Fig 1, 2, 3 and 5.

15. You should include a graphical abstract for clear understanding of the readers. You may modify Fig. 6 into graphical abstract after appropriate incorporation.

Reviewer #2: Overall, I think this study provides a valuable dataset for understanding the effects of iron deficiency and exogenous sucrose on the intermediates of chlorophyll biosynthesis in Malus halliana. However, there are several components of the manuscript that need to be improved before this paper is ready for publication, including grammatical issues that need to be addressed throughout the manuscript. Furthermore, with regard to grammar and structure, there are grammatically incorrect sentences or poorly written throughout the manuscript. I suggest that the authors work with an editor to revise their writing before resubmission. Few examples are given below. I suggest rejection of this article in this form.

In Introduction part:

Need to revise these sentence

Inhibition of any steps of Chl synthesis will cause reduced Chl content.

A study investigating the biochemical mechanism of Chl deficiency in pylm by examining Chl biosynthesis precursors in this mutant has shown that Chl biosynthesis is blocked in the mutant at the Chl a production step

Material and Method

In the material and method procedure regarding the isolation of RNA and preparation of cDNA was not discussed briefly.

In discussion part:

Fe deficiency stress upset the Chl biosynthesis balance in M. hupehensis, the Chl biosynthesis of M. halliana instead wasn’t blocked after Fe deficiency, which might be.

Besides, research has suggested that PbPPO1 genes might be involved in core browning under

modified atmosphere storage in ‘Yali’ pears[43].

Need to italic the technical name

And PPO gene expression level against Aspergillus tubingensis in table grapes was enhanced with trehalose

Need to revised this sentence

In this study, the PPOX gene was researched about the expression differences between two

apple rootstocks under Fe deficiency.

6. PLOS authors have the option to publish the peer review history of their article (what does this mean?). If published, this will include your full peer review and any attached files.

Reviewer #1: No

Reviewer #2: No

---

## [Author Response · Author response to Decision Letter 0]

29 Mar 2020

Dear Editors and Reviewers,

Thank you very much for your kind work and for the reviewers’ comments concerning our manuscript entitled “Effects of Iron Deficiency and Exogenous Sucrose on the Intermediates of Chlorophyll Biosynthesis in Malus halliana”. Those comments are all valuable and very helpful for revising and improving our paper, as well as the important guiding significance to our researches. We have studied comments carefully and have made correction which we hope meet with approval. We would like to revise manuscript if there are any requirements later. Revised portion are marked in red in the paper. The main corrections in the paper and the responses to the editor and reviewers’ comments are as following:

ACADEMIC EDITOR: 

1. Thank you for submitting your manuscript to PLOS ONE. I agree with the reviewers that English language needs to be checked thoroughly. Main claims of the paper are not properly placed in the context of previous literature. Authors did not treat the literature fairly. Introduction is too general.

Response: English language has been checked and modified by American Journal Experts (AJE)

To express well the main claims of paper, we have added more related literature and deleted some unnecessary literature in introduction parts. All changes have been marked in red in the manuscript.

Add (1) “The Chl biosynthesis pathway has many steps and involves various enzymes, and a blockade in one step will affect Chl biosynthesis and cause changes in leaf color[7].” in line 44-46.

(2) “Chl biosynthesis plays essential roles in photosynthesis and plant growth in response to environmental change[6].” in line 42-44.

(3) “The key regulatory sites of Chl synthesis are different for each crop under external stress.” in line 46-47.

(4) “A study of adzuki bean reported that the transformation of protoporphyrin IX (Proto IX) is blocked in Chl synthesis, causing etiolated seedlings[8]. ” in line 47-48.

(5) “Remarkably, Fe deficiency directly affected Chl synthesis[12].” in line 55-56.

(6) “Research in poplar revealed that Chl synthesis was inhibited under Fe-deficient conditions[13].” in line 56-57.

(7) “Spiller et al. attempted to study the effect of Fe on the Chl biosynthetic pathway, and the results indicated that Fe deficiency leads to the accumulation and excretion of intermediates in the tetrapyrrole biosynthetic pathway, particularly coproporphyrin[14].” in line 57-60.

(8) “Moreover, an investigation was initiated to locate possible sites where a deficiency of Fe might limit Chl synthesis of cowpea plants[2].” in line 60-61.

(9) “However, the responses of Chl biosynthesis to Fe deficiency stress and the key regulatory sites in M. halliana are still unknown.” in line 62-63.

(10) “However, little is known about how exogenous sucrose regulates the response of M. halliana to Fe deficiency through chlorophyll synthesis.” in line 71-73.

(11) “Studies of Chl biosynthesis have focused on various aspects[23, 24], whether biochemical[25, 26] or genetic[27, 28].” in line 74-75.

(12) “However, gaps remain in the knowledge of Chl biosynthesis and the related genes in apple rootstocks.” in line 75-76.

(13) “Therefore, it is important to elucidate the Chl biosynthetic molecular responses of M. halliana to Fe deficiency.” in line 76-77.

(14) “This study is original research work investigating the differences between two apple rootstocks under Fe deficiency at a molecular scale.” In line 80-81.

2. Response: We have updated statement about funding in my cover letter.

3. To enhance the reproducibility of your results, we recommend that if applicable you deposit your laboratory protocols in protocols.io, where a protocol can be assigned its own identifier (DOI) such that it can be cited independently in the future. For instructions see: http://journals.plos.org/plosone/s/submission-guidelines#loc-laboratory-protocols

Response: We have deposited laboratory protocols in protocols.io, and DOI is dx.doi.org/10.17504/protocols.io.beb6jare. 

Additional Editor Comments (if provided):

1. Thank you for submitting your manuscript to PLOS ONE. I agree with the reviewers that English language needs to be checked thoroughly. Main claims of the paper are not properly placed in the context of previous literature. Authors did not treat the literature fairly. Introduction is too general.

Response: English language has been checked and modified by American Journal Experts (AJE)

To express well the main claims of paper, we have added more related literature and deleted some unnecessary literature in introduction parts. All changes have been marked in red in the manuscript.

Add (1) “The Chl biosynthesis pathway has many steps and involves various enzymes, and a blockade in one step will affect Chl biosynthesis and cause changes in leaf color[7].” in line 44-46.

(2) “Chl biosynthesis plays essential roles in photosynthesis and plant growth in response to environmental change[6].” in line 42-44.

(3) “The key regulatory sites of Chl synthesis are different for each crop under external stress.” in line 46-47.

(4) “A study of adzuki bean reported that the transformation of protoporphyrin IX (Proto IX) is blocked in Chl synthesis, causing etiolated seedlings[8]. ” in line 47-48.

(5) “Remarkably, Fe deficiency directly affected Chl synthesis[12].” in line 55-56.

(6) “Research in poplar revealed that Chl synthesis was inhibited under Fe-deficient conditions[13].” in line 56-57.

(7) “Spiller et al. attempted to study the effect of Fe on the Chl biosynthetic pathway, and the results indicated that Fe deficiency leads to the accumulation and excretion of intermediates in the tetrapyrrole biosynthetic pathway, particularly coproporphyrin[14].” in line 57-60.

(8) “Moreover, an investigation was initiated to locate possible sites where a deficiency of Fe might limit Chl synthesis of cowpea plants[2].” in line 60-61.

(9) “However, the responses of Chl biosynthesis to Fe deficiency stress and the key regulatory sites in M. halliana are still unknown.” in line 62-63.

(10) “However, little is known about how exogenous sucrose regulates the response of M. halliana to Fe deficiency through chlorophyll synthesis.” in line 71-73.

(11) “Studies of Chl biosynthesis have focused on various aspects[23, 24], whether biochemical[25, 26] or genetic[27, 28].” in line 74-75.

(12) “However, gaps remain in the knowledge of Chl biosynthesis and the related genes in apple rootstocks.” in line 75-76.

(13) “Therefore, it is important to elucidate the Chl biosynthetic molecular responses of M. halliana to Fe deficiency.” in line 76-77.

(14) “This study is original research work investigating the differences between two apple rootstocks under Fe deficiency at a molecular scale.” In line 80-81.

Journal Requirements:

Response: We have revised the manuscript style according to style requirements. If the manuscript is still inappropriate, don't hesitate to contact me.

2. We noticed you have some minor occurrence of overlapping text with the following previous publication(s), which needs to be addressed: https://bmcplantbiol.biomedcentral.com/articles/10.1186/s12870-015-0699-7. In your revision ensure you cite all your sources (including your own works), and quote or rephrase any duplicated text outside the methods section. Further consideration is dependent on these concerns being addressed.

Response: We have addressed all overlapping text and cited all works sources in the introduction and discussion sections. 

3. In your Methods, please state the source of the seeds used in your study.

Response: We have stated the source of seeds and marked in red in line 85-86. 

4. While revising your submission, please upload your figure files to the Preflight Analysis and Conversion Engine (PACE) digital diagnostic tool, https://pacev2.apexcovantage.com/. PACE helps ensure that figures meet PLOS requirements.

Response: My uploaded figures have passed the PACE checks, and the PACE report signifies that the figure files need no adjustment.

Reviewer #1:

1. First you need to mention the common name in Title. You can mention scientific name in Abstract or Introduction in Parenthesis. Each scientific name has “L.” at the end according to nomenclature therefore, revise it thoroughly.

Response: We have submitted the article to AJE for language editing, and the title is recommended not to be changed. Additionally, we have revised thoroughly scientific name in abstract and introduction in parenthesis.

2. All abbreviations must be mentioned in full form at its first place. Mention all these abbreviations in footnotes of the first page.

Response: All abbreviations have been mentioned in full form at its first place, and all abbreviations have been mentioned in footnotes of the first page. 

3. In the last paragraph of Introduction you have described some results. You just need to mention the background of your work as well as the significance for scientific world. You should discuss results in discussion part.

Response: We have moved some results in the Introduction section to the discussion part. In the meantime, we have added the background of work and the significance for scientific world in the introduction parts (line 55-75). 

4. Insert line numbers in the manuscript for better review process.

Response: We have inserted line numbers in the manuscript.

5. Meaning is not clear “pylm”….is it Palm??

Response: We have confirmed again from the literature and determined that it is pylm and not Palm.

6. “nonphotosynthetic must be replaced with non-photosynthetic”

Response: This manuscript was edited and revised by the AJE. nonphotosynthetic cannot be replaced by non-photosynthetic.

7. You have mentioned “Increased Suc accumulation is required for regulating Fe-deficiency responses in plants, with auxins acting downstream in transmitting the Fe-deficiency signal”. How could you explain the downstreaming??

Response: This sentence comes from the 22nd reference to show that sucrose is required for regulating Fe-deficiency responses in plants (See 22nd reference).

8. “Exogenous Suc treatments”. Must mention Sucrose in full form in heading or subheadings. Similarly “Determination of Chl Precursors” as Determination of Chlorophyll Precursors.

Response: Sucrose in full form in heading or subheadings has been mentioned, and “Determination of Chl Precursors” has been replaced by “Determination of Chlorophyll Precursors”.

9. Caption of Fig 3. shows treatment codes as T1, T2 etc…all the punctuation must be appropriate like T3: Fe 21 d. Revise all codes in figure captions.

Response: We have revised all treatments codes in Fig 3 according to the above example.

10. You have used various abbreviations in your manuscript. You should also include Pn for Photosynthesis.

Response: We have added Pn (Photosynthesis) to the Abbreviation section.

11. “Stay green” must be replaced with greenness.

Response: “Stay green” has been replaced with “greenness”.

12. You must have to mention your main findings/conclusions under separate heading. Also mention your major outcomes in more elaborative way.

Response: We have mentioned main findings under separate heading and mentioned major outcomes in Results, Discussion and Conclusion parts.

13. Arrange the references alphabetically.

Response: Based on the PLoS ONE's style requirements, references should be arranged in the order in which they are inserted. So, we have arranged the references as requirements.

14. For the figures, you need to mention the x axis once for A, C, E and combine them, then mention x-axis scale once for B, D and F and combine them. Move Y-axis to right side of three figures (B, D, F). Similarly, move Y-axis on the left side of three figures (A, C, E). Combine all the six figures into single figure and enhance the dpi up to 500 for better understanding. Follow this procedure for Fig 1, 2, 3 and 5.

Response: We have generated the figures as required.

15. You should include a graphical abstract for clear understanding of the readers. You may modify Fig. 6 into graphical abstract after appropriate incorporation.

Response: We have tried to modified Fig 6 into graphical abstract. See Fig 6 and the Fig 6 legend (line 232-243) for the changes. We are willing to continue to modify when there are any requirements later.

Reviewer #2:

1. In Introduction part:

Need to revise these sentence

Inhibition of any steps of Chl synthesis will cause reduced Chl content.

A study investigating the biochemical mechanism of Chl deficiency in pylm by examining Chl biosynthesis precursors in this mutant has shown that Chl biosynthesis is blocked in the mutant at the Chl a production step

Response: We have revised and modified these sentences and the changes have been marked in red in the text.

2. Material and Method

In the material and method procedure regarding the isolation of RNA and preparation of cDNA was not discussed briefly.

Response: We have briefly added the isolation of RNA and preparation of cDNA in the material and method part.

3. In discussion part:

Fe deficiency stress upset the Chl biosynthesis balance in M. hupehensis, the Chl biosynthesis of M. halliana instead wasn’t blocked after Fe deficiency, which might be.

Besides, research has suggested that PbPPO1 genes might be involved in core browning under

modified atmosphere storage in ‘Yali’ pears[43].

Response: We have revised again these sentences as required, and the manuscript has been edited and modified by AJE. Changes are marked in red in the paper.

4. Need to italic the technical name 

And PPO gene expression level against Aspergillus tubingensis in table grapes was enhanced with trehalose.

Response: The technical name has been changed to italic and marked in red in the text.

5. Need to revised this sentence 

In this study, the PPOX gene was researched about the expression differences between two

apple rootstocks under Fe deficiency.

Response: We have deleted this sentence and restated in line 254-255.

We have added and revised details in the article according to your suggestions. Thank you again for improving our paper.

Best Regards,

Yours Sincerely 

Ai-xia Guo

Corresponding author:

Name: Yan-xiu WANG

E-mail: wangxy@gsau.edu.cn

---

## [Decision Letter · Decision Letter 1]

12 Apr 2020

PONE-D-19-35123R1

Effects of iron deficiency and exogenous sucrose on the intermediates of chlorophyll biosynthesis in Malus halliana

PLOS ONE

Dear Dr. Wang,

Thank you for submitting your manuscript to PLOS ONE. After careful consideration, we feel that it has merit but does not fully meet PLOS ONE’s publication criteria as it currently stands. Therefore, we invite you to submit a revised version of the manuscript that addresses the points raised during the review process.

ACADEMIC EDITOR: I really appreciate the efforts did by authors during revision, but according to the reviewer there are still some points which need to be corrected. 

We would appreciate receiving your revised manuscript by May 27 2020 11:59PM. To enhance the reproducibility of your results, we recommend that if applicable you deposit your laboratory protocols in protocols.io, where a protocol can be assigned its own identifier (DOI) such that it can be cited independently in the future. For instructions see: http://journals.plos.org/plosone/s/submission-guidelines#loc-laboratory-protocols

We look forward to receiving your revised manuscript.

Kind regards,

Basharat Ali, Ph.D

Academic Editor

PLOS ONE

Reviewers' comments:

Reviewer's Responses to Questions

**Comments to the Author**

1. If the authors have adequately addressed your comments raised in a previous round of review and you feel that this manuscript is now acceptable for publication, you may indicate that here to bypass the “Comments to the Author” section, enter your conflict of interest statement in the “Confidential to Editor” section, and submit your "Accept" recommendation.

Reviewer #1: All comments have been addressed

Reviewer #2: All comments have been addressed

2. Is the manuscript technically sound, and do the data support the conclusions?

Reviewer #1: Yes

Reviewer #2: Yes

3. Has the statistical analysis been performed appropriately and rigorously? 

Reviewer #1: Yes

Reviewer #2: Yes

4. Have the authors made all data underlying the findings in their manuscript fully available?

Reviewer #1: Yes

Reviewer #2: Yes

5. Is the manuscript presented in an intelligible fashion and written in standard English?

Reviewer #1: Yes

Reviewer #2: Yes

6. Review Comments to the Author

Reviewer #1: The authors did good efforts in revising the manuscript according to the comments. But at some points, I still feel that changes should be made. You should have to follow the below mentioned points before acceptance for publication.

1. Even you have revised your paper from an English Editor but you cannot mention this “Iron1” in Title. You can show corresponding author email with asterisk that will also show abbreviation list in footnotes which is usually followed in literature.

2. In the first revision I have mentioned this but I can’t find this in text. Each scientific name has “L.” at the end according to nomenclature therefore, revise it thoroughly.

3. In the first revision I have mentioned this but I can still find some related mistakes in the text. “Exogenous Suc treatments”. Must mention Sucrose in full form in heading or subheadings. Similarly “Determination of Chl Precursors” as Determination of Chlorophyll Precursors.

For example:

“Effects of Fe deficiency on the Chl precursor contents of M.halliana and M. hupehensis”

“Expression of genes involved in Chl biosynthesis under Fe deficiency”

“Regulation of the Fe deficiency phenotype by exogenous Suc”

4. For better understanding of the readers, you can move graphical abstract right after Abstract or Introduction before Materials and Methods. You can also quote this figure if there is any text relevant to this in the main body.

Reviewer #2: Authors have improved all the points very well. The manuscript has been revised well. Now paper is accepted for the publication in the journal.

7. PLOS authors have the option to publish the peer review history of their article (what does this mean?). If published, this will include your full peer review and any attached files.

Reviewer #1: Yes: Shahbaz Atta Tung

Reviewer #2: No

---

## [Author Response · Author response to Decision Letter 1]

15 Apr 2020

Dear Editors and Reviewers,

Thank you very much for your kind work and for the reviewers’ comments concerning our manuscript entitled “Effects of iron deficiency and exogenous sucrose on the intermediates of chlorophyll biosynthesis in Malus halliana”. Those comments are all valuable and very helpful for revising and improving our paper, as well as the important guiding significance to our researches. We have studied comments carefully and have made correction which we hope meet with approval. Revised portion are marked in red in the paper. The main corrections in the paper and the responses to the editor and reviewers’ comments are as following:

Reviewer #1

1. Even you have revised your paper from an English Editor but you cannot mention this “Iron1” in Title. You can show corresponding author email with asterisk that will also show abbreviation list in footnotes which is usually followed in literature.

Response: Thank you for your comments. We have revised “Iron1” in Title part, and we have showed corresponding author email with asterisk. All changes have been marked in red in the manuscript. If the manuscript is still inappropriate, don't hesitate to contact me.

2. In the first revision I have mentioned this but I can’t find this in text. Each scientific name has “L.” at the end according to nomenclature therefore, revise it thoroughly. 

Response: Thank you very much for your suggestion and patience. We have referred to the relevant literature (the 4th and 45th references in text). The scientific name of two apple rootstock in the text may not include "L.". Then we consulted the academic editor. Editor has indicated that the change that reviewer#1 suggested is not necessary. Combining relevant literature (the 4th and 45th references in text) and editorial reply, we decided not to review the scientific name in the full text of my manuscript. If you have any questions or concerns about my manuscript, please don't hesitate to reach out and let me know.

3. In the first revision I have mentioned this but I can still find some related mistakes in the text. “Exogenous Suc treatments”. Must mention Sucrose in full form in heading or subheadings. Similarly “Determination of Chl Precursors” as Determination of Chlorophyll Precursors.

For example:

“Effects of Fe deficiency on the Chl precursor contents of M. halliana and M. hupehensis”

“Expression of genes involved in Chl biosynthesis under Fe deficiency”

“Regulation of the Fe deficiency phenotype by exogenous Suc”

Response: Thank you very much for your suggestion and patience. We have revised related mistakes in heading or subheadings. All changes have been marked in red in lines 81, 104, 126, 141, 142, 163, 164, 181, 198, 217 and 218.

4. For better understanding of the readers, you can move graphical abstract right after Abstract or Introduction before Materials and Methods. You can also quote this figure if there is any text relevant to this in the main body.

Response: Thank you very much for your suggestions. We can move graphical abstract right after Abstract or Introduction before Materials and Methods, but I think it's more understandable for the authors and readers without moving the graphical abstract. Additionally, this figure is a schematic model of key genes involved in chlorophyll biosynthesis of apple rootstock during iron deficiency and sucrose application. We haven't found a consistent graphics yet. Therefore, we didn’t quote this figure in the main body.

We have revised some details in the article according to your suggestions. Thank you again for improving our paper.

Best Regards,

Yours Sincerely 

Ai-xia Guo

Corresponding author:

Name: Yan-xiu WANG

E-mail: wangxy@gsau.edu.cn

---

## [Decision Letter · Decision Letter 2]

21 Apr 2020

Effects of iron deficiency and exogenous sucrose on the intermediates of chlorophyll biosynthesis in Malus halliana

PONE-D-19-35123R2

Dear Dr. Wang,

We are pleased to inform you that your manuscript has been judged scientifically suitable for publication and will be formally accepted for publication once it complies with all outstanding technical requirements.

With kind regards,

Basharat Ali, Ph.D

Academic Editor

PLOS ONE

Additional Editor Comments (optional):

Reviewers' comments:

Reviewer's Responses to Questions

**Comments to the Author**

1. If the authors have adequately addressed your comments raised in a previous round of review and you feel that this manuscript is now acceptable for publication, you may indicate that here to bypass the “Comments to the Author” section, enter your conflict of interest statement in the “Confidential to Editor” section, and submit your "Accept" recommendation.

Reviewer #1: All comments have been addressed

2. Is the manuscript technically sound, and do the data support the conclusions?

Reviewer #1: Yes

3. Has the statistical analysis been performed appropriately and rigorously? 

Reviewer #1: Yes

4. Have the authors made all data underlying the findings in their manuscript fully available?

Reviewer #1: Yes

5. Is the manuscript presented in an intelligible fashion and written in standard English?

Reviewer #1: Yes

6. Review Comments to the Author

Reviewer #1: The authors have revised all the suggested changes positively in the revised version. The manuscript is suitable for publication in PLOS ONE.

7. PLOS authors have the option to publish the peer review history of their article (what does this mean?). If published, this will include your full peer review and any attached files.

Reviewer #1: Yes: Shahbaz Atta Tung

---

## [Editor Report · Acceptance letter]

24 Apr 2020

PONE-D-19-35123R2 

Effects of iron deficiency and exogenous sucrose on the intermediates of chlorophyll biosynthesis in *Malus halliana*

Dear Dr. Wang:

I am pleased to inform you that your manuscript has been deemed suitable for publication in PLOS ONE. Congratulations! Your manuscript is now with our production department. 

With kind regards,

on behalf of

Dr. Basharat Ali 

Academic Editor

PLOS ONE